# Adipocyte-Derived Stem Cells in the Treatment of Spinal Cord Injuries in Animal Models: A Systematic Review

**DOI:** 10.3390/ijms26199330

**Published:** 2025-09-24

**Authors:** Faraz Jamil, Taha Ahmed, Hamzah Iqbal, Antonia Vogt, Wasim Khan

**Affiliations:** 1School of Clinical Medicine, Cambridge University, Cambridge CB2 0SP, UK; fj275@cam.ac.uk (F.J.); ta504@cam.ac.uk (T.A.); hai21@cam.ac.uk (H.I.); 2Division of Trauma and Orthopaedic Surgery, Addenbrooke’s Hospital, University of Cambridge, Cambridge CB2 0QQ, UK; av591@medschl.cam.ac.uk

**Keywords:** spinal cord injury (SCI), adipocyte-derived mesenchymal stem cell (ADSC), locomotor recovery

## Abstract

Spinal cord injury (SCI) represents a burdensome and currently incurable condition which affects over 20 million patients globally. Adipocyte-derived mesenchymal stem cell (ADSC) therapy may constitute a valuable strategy in treating this condition, owing to their unique cellular characteristics and beneficial effects on functional recovery. This PRISMA (Preferred Reporting Items for Systematic Reviews) review aims to assess whether ADSC therapy is a viable strategy for treating SCI in animal models. We identified a total of 1561 studies after performing a search of four databases, including PubMed, Web of Science, Scopus and Medline. After applying inclusion and exclusion criteria, we identified a total of 16 articles that were reviewed, assessed and reported in our study. General characteristics of these studies, results of stem cell characterisation, SCI induction protocols, locomotor recovery and bladder function following SCI, were investigated as part of our analysis. Fifteen studies suggested that ADSC therapy has a beneficial effect on motor recovery following SCI. The evidence base regarding adjuvant therapies was, however, variable. Further investigations into the mechanisms that underly recovery following ADSC therapy, and potential adjuvants which could enhance these effects, should follow the outcomes of this systematic review. In turn, this would help expand the treatment options available to SCI patients.

## 1. Introduction

### 1.1. Spinal Cord Injury [SCI]

SCI is defined as a multidimensional disorder arising from direct or indirect trauma to the spinal cord, leading to a sequela of autonomic, motor and sensory dysfunction and disability [1]. It is a significant cause of morbidity and mortality globally, with social costs that exceed rational quantification. An estimated 20 million people suffer from SCI globally, and there are approximately 0.9 million new incidents per annum [2]. The most common cause of an SCI is high energy trauma, particularly following motor vehicle accidents (MVAs) and falls. A significant proportion of SCI cases are also caused by sports-related trauma [3]. Most alarmingly, SCI appears to be increasing in incidence, with males and the elderly disproportionally affected compared to other groups [3].

Outcomes following SCI may vary significantly, being affected by several factors. Indeed, the long-term outcomes in vulnerable (socioeconomically or otherwise) individuals was seen to be consistently worse across metrics [4]. This included higher rates of mortality and disability, alongside an increased risk of chronic pain. Numerous mechanisms may explain this association, with one plausible explanation being differences in access to physiotherapy. This, in turn, has significant impacts on quality of life, further augmenting the impact of SCI on the individual. It is hence evident as that the availability of treatment is a significant limitation on SCI patients, which is particularly alarming given the increased incidence of this condition in developing countries [2]. Taken together, there is a strong case for the development of accessible therapeutics which help reduce the overall disease burden of SCI, both on an individual and a global scale.

An SCI occurs secondary to direct or indirect trauma to the spinal cord, which may occur from either external forces or through bony fragments from fractured vertebrae, respectively. It is a result of the vertebral column failing to adequately absorb high energy forces and protect the spinal cord. The nature of the impact force, including its magnitude and component vectors, and the level at which the subsequent SCI occurs, all affect the outcomes that follow.

The cervical spine is the most common site of injury, representing 43.9–61.5% of SCIs [5]. Furthermore, patients with C1–C4 level injuries display shorter post-injury life expectancies than those with C5–C8 injuries [6]. This follows from the structure of the spinal cord itself, with more proximal injuries leading to a greater degree of disability and disruption. This underlines the importance of subcategorising SCIs according to their exact pathology. It is therefore important that the animal models used in the study of SCI accurately simulate the patterns of injury observed in humans.

### 1.2. Phases of SCI

A spinal cord injury may be divided into four main pathological phases [1]:-Acute phase (2 h–2 days): Characterised by hallmarks of acute injury, namely inflammation, haemorrhage and oedema. Other features of note are vascular disruption, cellular necrosis and axonal injury. There is also evidence that free radical generation further contributes to cellular damage in the acute phase [7].-Subacute phase (2 days–2 weeks): Phagocytosis of debris begins to occur; axonal growth is initiated. At this stage, astrocytes proliferate and migrate to the site of injury [8]. However, phenotypic changes lead to glial scarring which may later impede neural regeneration in the chronic phase.-Intermediate phase (2 weeks–6 months): Maturation of scarring and axonal sprouting begins.-Chronic phase (6 months onwards): Formation of chronic scar tissue which may impede axonal growth [9]. Wallerian degeneration also occurs at this stage, and long-term sequela such as chronic pain and motor dysfunction begin to develop.

Therapeutic approaches would therefore aim to mitigate the damage which occurs during early phases of SCI; these early stages form the basis of long-term spinal cord dysfunction. Stem cell therapy has the potential to directly counteract the pathological mechanisms at play during these initial phases.

### 1.3. Mesenchymal Stem Cells (MSCs)

MSCs are stromal cells which possess the ability to both self-renew and exhibit multi-lineage differentiation [10]. MSCs were first isolated in the 1970s from bone marrow, displaying a unique ability to differentiate into fibroblastic colony-forming cells [11]. MSCs may be derived from a diverse array of tissues, and through a variety of methods. This review mainly concerns the derivation of MSCs from adipose tissue; so-called adipose derived mesenchymal stem cells (ADSCs).

MSCs display several defining characteristics. They are typically triple positive for the cell surface markers CD73, CD90 and CD105, and negative for CD34 [12]. However, this classification is contentious given evidence that not all populations of stem cells display these precise characteristics [13]. This is unsurprising given the sheer heterogeneity with which MSCs are derived, cultured and experimentally programmed. There is an argument that the term MSC is, above all, a general descriptor for these unique cell types. Nonetheless, the International Society for Cell and Gene Therapy has set a minimum criterion that defines MSCs [14].

MSCs possess the ability to differentiate into a plethora of cell lineages. This includes neuronal cell types, osteocytes, myocytes and hepatocytes, among others. They are highly versatile cells which possess immense therapeutic potential in treating SCI. This derives from two potential mechanisms. The first is their secretory profile; MSCs have a propensity to secrete trophic molecules in accordance with their local environment; this may support growth mechanisms whilst also limiting deleterious processes such as oxidative stress [15]. Secondly, there is potential to directly differentiate MSCs into neural tissue; this could be utilised as a form of neural replacement therapy [16]. MSC transplantation has demonstrated benefits in preventing both neural inflammation and ischaemia.

#### 1.3.1. ADSCs

Adipocyte-derived stem cells are a unique form of MSC. Of note is their availability in virtually all individuals and their relative ease of access, compared to stem cell harvesting from bone marrow, for instance. This presents an opportunity for autologous therapy, a strategy that has been clinically validated in humans, with profound benefits displayed [17].

ADSCs were first identified in 2002 by Zuk and colleagues [18]. Since their initial discovery, their therapeutic potential has been further elucidated. Of relevance to SCI, these cells have been shown to differentiate into neural cell types [19]. As discussed previously, this presents the potential of autologous ADSC isolation and differentiation into neural tissue, thereby providing a supply of cells to replace damaged neural tissue. This review primarily concerns SCI therapy that uses undifferentiated ADSCs, which have unique secretory profiles. This includes the secretion of lactate which appears to inhibit M1 macrophage-induced inflammation [20]. The potential impact of this upon the early phases of SCI are numerous, given the central contribution of inflammation towards pathology. An important consideration is the exact stage and passage at which ADSCs are transplanted, as this may alter their secretory profile and ergo therapeutic profile. At the most basic level, ADSC therapy is a clinically validated strategy, with over 130 clinical trials ongoing, several with encouraging results [21]. The primary concern is then validating their use in the treatment of SCI, as well as further elucidating mechanisms to optimise any potential benefits.

#### 1.3.2. Animal Models Relevant to SCI Research

There are a wide range of animal models used in the study of SCI. Most of these are rodent models, but there are also several canine models in use, as well as case reports of animals such as domestic ferrets [22]. Our study includes 15 rat models and 1 dog model. Of these, eight studies use human-derived ADSCs, and the rest are species-matched (i.e., dog canine ADSCs in a dog model). Locomotor recovery was assessed using the Basso, Beattie and Bresnahan (BBB) scoring system, and improvements in BBB scores are interpreted as evidence of locomotor recovery.

Overall, the most robust evidence is likely to derive from rodent models owing to the ease of availability, and the well-established methods of inducing SCI that exist. These methods encompass modalities such as piston impacts, weight drops and direct cutting [23]. It is, however, necessary that the translational relevance of these methods is appraised, as will be elaborated in the discussion section. Encouragingly, translational anatomy studies suggest that, despite their smaller stature, mice represent a reasonably accurate physiological system in modelling humans [24]. Nonetheless, efforts to further enhance the accuracy of preclinical SCI models are likely to be with merit in ensuring translational success.

This systematic review aims to assess the evidence that supports the use of ADSC therapy for treating SCI in animal models, evaluating the methodological quality of studies obtained. Moreover, this review will examine the studies in terms of stem cell characterisation and SCI models, drawing comparisons between protocols. This review is the first of its kind in assessing evidence for the use of undifferentiated ADSCs in SCI and highlights key weaknesses which need to be addressed in order for the field to progress. Overall, this study will report all the available scientific literature, synthesising it into a comprehensive review. A critical assessment of the field will be made, with potential improvements and future directions suggested.

## 2. Materials and Methods

### 2.1. Search Strategy

A systematic review of the literature was performed according to the Preferred Reporting Items for Systematic Reviews (PRISMA) guidelines [25]. A search of the literature was carried out across four databases: PubMed, Web of Science, Scopus and Medline. Searches were performed during the first two weeks of November 2024 (4 November 2024–15 November 2024). Our search strategy encompassed the following terms and all their possible endings using the truncation operator “*”: “adipose”, “adipocyte”, “stem cells”, “MSCs”, “spinal cord injury” and “spinal cord injuries.” These search terms were combined by using Boolean operators “AND” and “OR”. The truncation operator “*” was used to maximise the number of articles obtained by our searches; articles would have been missed if alternative word endings had not been considered. One example search used was as follows: “adipo *” AND “stem cells” AND “spinal cord injur *”, illustrating our search strategy in practice.

In total, 280 studies were extracted from PubMed, 538 from Web of Science, 537 from Scopus and 206 from Medline. Searches were carried out by F.J., H.I. and T.A. during the first week of December 2024.

We identified 707 duplicates from these studies and removed them accordingly. Following the application of inclusion and exclusion criteria based on study titles and abstracts, 794 studies were excluded. A total of 60 studies were then assessed as per their full-text content against the inclusion and exclusion criteria listed below. Figure 1 explains the process we undertook, which culminated in the extraction of data from 16 final papers.

Searches were carried out by F.J., T.A. and H.I., and two authors (F.J. and T.A.) independently screened titled abstracts. Likewise, two authors independently carried out full-text screening of articles (F.J. and H.I.). In all instances of dispute, a mutual conclusion was reached following discussion between researchers.

The study was registered on the PROSPERO (International Prospective Register of Systematic Reviews) database under the registration number CRD420250607715.

### 2.2. Inclusion Criteria

Studies carried out in animal models;Studies where the primary intervention is the administration of ADSCs;Studies displaying stem cell characterisation results;Studies that utilise the [BBB] scale to measure locomotor outcomes;English language.

### 2.3. Exclusion Criteria

Studies with a total *n* value less than 10;Studies investigating conditions other than SCI;Studies not carried out in animal models;Studies lacking adequate control groups;Systematic reviews, the literature reviews and letters to editors;Studies not available in English.

### 2.4. Data Extraction

Sixteen articles were utilised in the data extraction process, with extracted data being displayed in an Excel spreadsheet. The data are presented in the four tables below, listing study references, general characteristics of studies, the SCI injury models used, locomotor recovery outcomes and outcomes related to bladder function.

### 2.5. Quality Assessment

Quality assessment of studies included in the data extracted was performed utilising “SYRCLE’s risk of bias assessment tool” [26]. Any disputes were resolved by discussion between researchers.

### 2.6. BBB Scoring System

The BBB scoring system was used throughout the studies to assess locomotor recovery. It is a 21-point scale that corresponds to sequential stages of motor recovery. Researchers in this sample of studies based scoring upon the observation of hindlimb movements and locomotor function. BBB scores were taken at set timepoints following the induction of SCI and the time courses of locomotor recovery plotted to allow comparison between groups.

## 3. Results

### 3.1. General Study Characteristics

This systematic review contains 16 studies, as illustrated in Figure 1. Studies that met inclusion/exclusion criteria were dated between 2009 and 2024 [27,28]. Most of these studies possess a broadly similar experimental framework, administering ADSCs to animals with experimentally induced SCI. All studies monitored locomotor recovery following this intervention. However, key differences between studies were evident in the methods used to induce SCI and whether adjuvant therapies were included. ADSCs were sourced from a broad range of samples: eight studies from human adipose tissue, seven from rat adipose tissue, and one study utilised canine adipose tissue. Significant variance in sample size was further evident; the smallest study had just 11 animals while the largest had 108 [29,30]. In all cases, subjects were split across several experimental groups including controls. One study was distinct in its experimental procedure, excluding rats that maintained BBB scores above 3 following SCI [31]. Moreover, ADSC culture conditions were broadly similar across studies. Culture media comprised mixtures of the following substances: DMEM (Dulbecco’s Modified Eagle Medium), Ham’s F-12, 10% FBS (Foetal Bovine Serum) with occasional supplementation of 1% penicillin/streptomycin. Stem cell incubation conditions (temperature and gaseous composition) were also broadly homogenous. ADSC isolation relied on ADSC pellets being obtained by centrifuging of samples. These pellets were then suspended into culture media, with non-adherent cells washed off and the medium replaced at regular intervals. Overall, the 16 studies are largely homogenous by design, (Table 1).

### 3.2. Characterisation of Adipocyte-Derived Stem Cells

All studies characterised stem cells obtained, with 13 out of 16 utilising flow cytometry. In the other three studies, immunocytochemistry [27], immunophenotyping [38] and immunofluorescence [42] were the methods utilised, respectively. Across studies, the most expressed cell surface marker was CD90 (12 out of 16 studies). Other commonly expressed markers included CD44 (eight studies), CD105 (seven studies), CD73 (seven studies) and CD29 (six studies). CD54 and CD13 were only found to be expressed in one study, respectively [32,41]. Expression patterning of MSCs has been independently validated in other studies [43]. Beyond CD markers, one study found ADSCs to express GFAP and beta-3 tubulin while another found that cells were positive for vimentin and BDNF [30,33]. Immunocytochemistry characterisation process revealed that ADSCs were triple positive for nestin, vimentin and beta-3 tubulin [27]. This study found that Ki67 (a protein used to measure rates of cancer cell division) was expressed in 57.22% ± 5.76% of the ADSCs characterised. Several studies noted differences in the expression patterns of surface markers as passage number increased. One study found that nestin expression decreased significantly as passage number increased [33]. Another paper compared stem cells from the seventh generation to the first, finding that the former lacked MHC-1 and CD34 expression [29]. Studies were non-homogenous in terms of stem cell stage at the point of characterisation, including whether cells were differentiated or not. There did not, however, appear to be any differences in the broad patterns of surface marker expression despite different stem cell sources (Table 2):

### 3.3. SCI Injury Models

All studies included in this review experimentally induced SCI in subject animals. Most studies (13 of 16) began with laminectomy of some form to expose the spinal cord. Following this, a variety of mechanisms were used to injure the exposed spinal cord. These methods ranged from contusion injuries [most common] induced by piston devices to direct lacerations with micro scissors [30,34]. Studies were highly heterogeneous in this regard. Most studies examined high thoracic level SCIs, and there was only one study that induced a true lumbar SCI [29]. Uniquely, one study induced a multi-level infarction injury through an aortic occlusion/reperfusion method [28]. There were no cervical SCI models in this sample. Furthermore, a variety of mechanical devices were used to induce SCI. Where specified, weights and pistons generally weighed 10 g, but other physical parameters such as impact distance differed notably. Likewise, spinal cord dwell times were inconsistent between studies using compressive methods such as clips and clamps [41,42]. One study further utilised an Infinite Horizon impactor device to induce spinal cord contusion [37]. SCI induction methods were highly heterogeneous in this sample of studies (Table 3).

### 3.4. BBB Scores

All 16 studies measured locomotor recovery via the BBB scoring system [44]. One study modified this into an Olby scale suitable for canines [29]. Fifteen studies found statistically significant differences between the BBB scores of ADSC-treated animals compared to controls. Control groups often received either PBS or physiological saline administered as placebo. Furthermore, one study compared the effects of ADSC therapy during the acute versus hyperacute phases of SCI. It was found that ADSC therapy led to significant improvements in motor function across both phases when compared to controls [32]. Another study also found that ADSC therapy caused significant improvements in motor function that BMSC therapy alone did not [30]. Eight studies further investigated potential adjuvant therapies. ChABC/ADSC combination therapy led to significantly improved BBB scores compared to ADSC groups [34]. This finding was further replicated in another study [37]. GCSF/ADSC combination therapy did not lead to statistically significant differences in BBB scores compared to ADSC groups [36]. One study supplemented ADSC therapy with exercise; ADSC/exercise groups had significantly higher BBB scores compared to all other groups [35]. The results suggest ADSC therapy to improve BBB scoring following SCI induction (Table 4).

### 3.5. Secondary Bladder Functional Outcomes

Two studies investigated the effects of ADSC therapy upon bladder function [27,38]. One study found that ADSC therapy led to a statistically significant increase in urinary continence [27]. Over 60% of animals cell-treated groups recovered continence while no control group animals recovered. There was, however, no statistical significance in bladder recovery between ADSC and ADSC/MPSS groups. There was a time lag observed between cellular transplant and measurable recovery [average of 9 days from the second transplant]. The second study compared the histological profiles of bladder tissue from ADSC and control groups, stained with HE and Masson [38]. Histological architecture was better preserved in ADSC-treated groups, which displayed less of the features characteristic of a bladder following SCI. Statistical significance was not quantified, and functional outcomes were not directly measured. These studies provide some evidence that ADSC therapy limits bladder dysfunction following SCI (Table 5).

### 3.6. Quality of Included Studies

A modified SYRCLE’s Risk of Bias (RoB) tool was used to grade each study using the 10 categories below [26]. A scale of 1–3 was utilised to rate each category, with an average being calculated to assign a final rating for each study. A score of 1 denotes low risk of bias, 2 unclear risks, and 3 a high risk of bias. Overall, three studies were definitely low risk. This was largely driven by a lack of reporting in terms of sequence generation used to allocate animals randomly across experimental groups. There were some concerns in assessing baseline characteristics of experimental groups in two studies, and attrition bias in three studies. Overall, none of the studies were definitely high risk, with thirteen of sixteen studies falling into the category of unclear overall risk of bias. All sixteen studies included in this study were of high quality with no major overall concerns of bias (Table 6).

## 4. Discussion

### 4.1. Stem Cells in Curing SCI

SCI is a currently incurable condition which consists of several phases. ADSC therapy may have potential to mitigate pathological mechanisms both in the early stages, and chronic stages of SCI. Notably, the secretome of ADSCs appears to limit the mechanisms which contribute to SCI severity [45]. Hence, the question at hand is whether direct ADSC transplantation represents a viable strategy in treating SCI. The design of the 16 studies is largely homogenous; a key distinction is the source of stem cells. These can be broadly divided into human- and animal-sourced ADSCs. This is valuable in drawing comparisons between methods of stem cell therapy. The relative consistency in terms of culture conditions further adds to the weight of these results. This trend of methodological consistency should continue. Further improvements to experimental procedures, such as standardising sample sizes to be generally large would further improve the quality of evidence obtained. Additionally, alternative sources of ADSCs could be explored; recent developments surrounding the use of porcine ADSCs in treating canine osteoarthritis suggest potential merits in further exploring xenotransplantation [46]. It would also be fascinating to explore the behaviour of ADSCs in a human-like environment, for example, in novel spinal cord on a chip model [47]. This would further elucidate the influence of local environmental factors on ADSC behaviour and inform the choice of culture conditions.

### 4.2. ADSC Characterisation

All studies included in our synthesis characterised ADSCs obtained; the majority achieved this using flow cytometry. ADSCs were screened for markers of stemness in accordance with ISCT guidelines [14]. Most stem cells obtained displayed characteristics consistent with the definition of a mesenchymal stem cell. However, there was a degree of heterogeneity in both the results of characterisation and underlying methodology. For example, atypical markers such as CD166 were found to be expressed in some ADSC populations. Likewise, studies differed in the stage at which they characterised cells. Some studies further assessed for markers such as vimentin, which are not included in ICST guidelines. CD105 was also more commonly expressed in human-derived ADSCs compared to rat-derived ADSCs. This dual heterogeneity creates a challenge in defining ADSC therapy and, consequently, assessing its therapeutic value. We suggest the following changes: (A) methodological consistency in the markers used to screen ADSCs for stemness and (B) a unified definition of what constitutes a stem cell, considering recent disputes regarding the validity of existing markers [13]. These changes would improve the consistency between studies and enable more robust comparisons to be made, minimising the number of confounding variables therein.

### 4.3. Quality of Experimental Models

ADSC induction protocols involved exposure via laminectomy, followed by direct trauma through several methods including weight drops and balloon inflation. It is currently unclear how relevant these methods of injury are to human SCI. These injuries certainly differ to the mechanisms of most human SCIs, removing the role of the vertebral column in force absorption. Moreover, direct trauma to an exposed human spinal cord is a rare cause of SCI; more commonly, SCI occurs due to the vertebral column being overcome by high forces and possible secondary injuries from bone fragmentation. Potential extensions could include models that simulate RTA-like injuries in animal models or falls from heights.

The experimental models primarily concerned thoracic SCI, with one true lumbar SCI model [29]. This is in direct contrast to human SCI patterns, where cervical SCI is predominant [5]. This limits the translational relevance of these results as SCI outcomes are significantly impacted by the level, particularly with regard to complete injuries. Disability occurs below the level of injury, and it is likely that the current rodent models fail to recapitulate the impairments that humans face following SCIs at higher spinal levels. This likely explains why the BBB scale itself measures hindlimb locomotor function, as the level of injury in most experiments is unlikely to disable forelimb function to the same extent. Cervical rat models do exist, and we suggest ADSC therapy is tested in such models to further strengthen the current evidence for their efficacy [48]. Another aspect to consider is using aged rodents to simulate frailty, given the incidence of SCIs in elderly humans following falls. Examining the efficacy of ADSC therapy stratified by the age of mice models would be fascinating and highly relevant to the translational capacity of these results.

Overall, current animal models overlook several aspects of human SCI incidence patterns and limit the translational relevance of their results. Nonetheless, they appear to provide some basic proof of concept, having been translated to successful human studies [17]. Whether these results ultimately lead to widespread clinical use is to be seen, ensuring translation will require improving current models to most accurately simulate human SCI patterns.

### 4.4. ADSC Therapy and Its Effects on Locomotor Recovery

Fifteen of sixteen studies found significant differences between post-SCI motor recovery when comparing ADSC groups to controls. In all cases, motor function was measured via the BBB scoring system, which assesses hindlimb function. Differences were usually observed sometime after stem cells were administered, generally beyond a full week. This reveals temporal dynamics related to stem cell therapy for SCI; with functional recovery occurring beyond the acute phase of SCI.

We suggest two potential explanations for this: one could be the time taken for ADSCs to migrate and begin producing trophic cytokines, while another could be the time taken for spinal cord recovery to translate to observable functional improvements. In either case, these studies suggest that ADSC therapy improves motor recovery following SCI, the pathways that underly this are partially elucidated. One study provides compelling evidence that TGF plays some role, with significant differences observed between ADSC-only groups compared to ADSC + TGF-1R inhibitor groups [41]. Another study suggests that ADSC therapy is enhanced when delivered in a three-dimensional cell mass [42]. We conclude that the fine parameters of ADSC therapy and how best to optimise its use clearly extends beyond basic cell transplantation. There is likely a range of factors, including cytochemical and spatial, which influence the overall efficacy of this therapy. The results from these studies provide strong evidence for the efficacy of ADSC therapy alone, but the landscape regarding adjuvant therapy is more contentious.

The efficacy of ADSC therapy may also vary with their stage at the time of transplant. Undifferentiated and passage 2 ADSC groups were found to have significantly higher BBB scores than passage 1 groups [27]. The time of administration relative to SCI was variable across studies. A consistent pattern of lag was identified; ADSC therapy took several days (sometimes weeks) to promote locomotor recovery. The current working model we suggest is as follows: ADSC therapy is effective beyond the acute phase of SCI but alone may be insufficient to ensure full recovery. Bridging therapy may be necessary (discussed further in Section 4.6) to do so. Additionally, we suggest further studies to investigate optimisation strategies, such as alternate delivery modules, and adjuvant therapies.

The effect of age on the efficacy of ADSC therapy needs to be further investigated to strengthen the translational relevance of these results, especially since a large number of SCI injuries in humans occur in elderly populations. Moreover, examining the results stratified by species, including cell origin, suggests broadly consistent findings. However, this is limited by the focus of reviewed studies on smaller organism models, which are limited in certain aspects of translational relevance. The variation in locomotor testing protocols must also be highlighted as a potential source of bias which may limit the strength of these results.

Overall, ADSC therapy appears to present some beneficial effects in improving post-SCI motor recovery, but further understanding of underlying mechanisms and therapeutic approaches is necessary before successful translation to the clinic.

### 4.5. Other Effect Measures

Two studies investigated the benefits of ADSC therapy on bladder recovery following SCI, both found evidence of therapeutic utility [27,38]. Taken together, these studies suggest a model where ADSC therapy may promote cellular recovery (evidenced by histological observations) that may help restore bladder functions such as urinary continence. This is encouraging, suggesting that ADSC therapy has benefits that extend beyond motor recovery alone. Autonomic dysregulation is a significant cause of morbidity and mortality following SCI, particularly above T6 level [49]. We suggest further studies investigating the utility of SCI therapy on minimising autonomic disruption, creating the potential for ADSC to aid nervous recovery at several levels. Autonomic recovery, although crucial in ensuring quality of life for SCI patients, is often unappreciated in studies. It is imperative that the field shifts towards assessing a broader range of autonomic outcomes in animal models, including cardiac and bowel function, for example. This will greatly improve the translational strength of results obtained.

### 4.6. Adjuvant Therapies

Several studies investigated adjuvant therapies alongside ADSCs, with varying results. Overall, some form of bridging therapy may have utility given the time taken for ADSC therapy to yield observable benefits. For instance, one study found ChABC monotherapy to promote statistically significant motor recovery two weeks faster than ADSC therapy [34]. Another study found ADSC therapy to only be effective when combined with exercise therapy [35]. Taken together, these studies provide evidence that ADSC therapy alone may not be optimal or even sufficient to induce motor recovery following SCI. However, a clear rationale for adjuvant therapy use is still not available. No additive benefit was found with GCSF adjuvant despite its cellular effects being theoretically useful in functional recovery [32]. The field requires further research to optimise the parameters of ADSC therapy, including adjuvant and delivery approaches. It would be fascinating to investigate which combinations of adjuvants display synergy, leveraging the current understanding of molecular mechanisms and pathways related to SCI.

It is fascinating to note that certain treatment combinations yield additive effects while others do not. This could be related to the underlying molecular pathways of studies, and whether they meaningfully interact. Agonistic pathways may underly combination therapies which yield additive effects, such as ADSC combined with a self-assembling peptide [39]. The exact details behind these effects are partially elucidated, and far more molecular research is needed to fully understand the basis of observed results. We draw particular attention to the study which suggests that exercise permits ADSC therapy to be effective at all [35]. Studies such as this highlight further knowledge gaps within the field, and we strongly suggest future endeavours aimed at dissecting molecular pathways involved in ADSC therapy. This is a vital step toward further validating these results, and optimising ADSC therapy for human use.

### 4.7. Limitations of Studies

The results of these studies, while encouraging, are limited by methodological heterogeneity between studies. Varying sources of ADSCs were utilised, with the genetic profiles of cells differing between species. We draw attention to the subset of eight studies which utilised human-derived ADSCs, all of which found significant benefits with therapy. We encourage further studies to utilise human ADSCs as the standard treatment format, as this better mimic what is likely to translate to human therapy. Nonetheless, differences between exact culture approaches and surface expression may limit comparisons between studies. We also highlight that our language restriction criterion, focusing on English-language studies, may have restricted results obtained.

There was also variance in the methods of injury, with a variety of mechanical devices being utilised. This created challenges when comparing between studies, as it is uncertain how the method of injury influences SCI pathology. These animal studies are also limited by the fact that they rely on exposing the spinal cord prior to injury, which contrasts with human patterns of injury that occur within the vertebral column. There is also a dire need for more cervical models in the literature, given the relevance of c-spine injuries to human SCI patterns. There are limitations of extrapolating thoracic-derived results, as these studies present, to human patterns of cervical SCI. To further strengthen the evidence presented, we implore researchers to standardise the animal models used, ideally focusing on larger organisms closer to humans, and shifting toward cervical SCI studies.

The BBB scoring system is a standardised scoring scale for locomotor function utilised in rodents, and modified versions have been created for use in canines [29]. The reliance on a singular scoring scale, while facilitating cross-study comparisons, may create potential for bias in results. Firstly, publication bias is a possibility we must mention, as is the issue of transparency in outcome assessment approaches. This is relevant as several studies reviewed did not specify whether reviewers were blinded, and the exact methods of blinding utilised. It is imperative that the data transparency within the field is improved in future, particularly providing raw results of figures used to create BBB scoring graphs, which would aid independent statistical analysis.

The literature provides some encouraging evidence for ADSC therapy in animal models following SCI but is limited by the heterogeneity that exists between studies and key translational limitations of current models. We implore researchers to explore more relevant methods of injury, animal models and locations of SCI to further strengthen the evidence base of this therapeutic approach.

## 5. Conclusions

This systematic review demonstrates some evidence that ADSC therapy following SCI may improve locomotor recovery in animal models. While the majority of studies provide evidence for this model alone, the landscape regarding adjuvant approaches is more contentious. In addition, most studies support the idea that ADSC therapy has a lag period, in which a bridging therapy may be useful. There was also some limited evidence to suggest that ADSC therapy may limit autonomic dysfunction following SCI. These results were limited by the heterogeneity between studies and key weaknesses in the study design, which limits translational relevance. Further investigations into the mechanisms of ADSC therapy and optimal adjuvant approaches should follow the outcomes of this systematic review. We draw particular attention to the molecular mechanisms of adjuvant therapy and the need for further research in this area. The results from this study should also contribute to the wider body of evidence supporting the trial of ADSC therapy in human patients suffering from SCI, following validation in more translationally relevant experimental models. Translating ADSC therapy to patients suffering from SCI will depend heavily on addressing the key weaknesses in the current literature, thereby forming a stronger proof of concept to justify large-scale human trials.

SCI continues to be a debilitating and burdensome condition, both at the individual and societal level. ADSC therapy, while still at an early stage, could pave the way for a cure for the profound disability that follows from an SCI. Recent developments in technology, from transcriptomics to advanced nanoparticle delivery devices, could help scientists cure SCI once and for all if the weaknesses of the current literature are addressed.

## Figures and Tables

**Figure 1 ijms-26-09330-f001:**
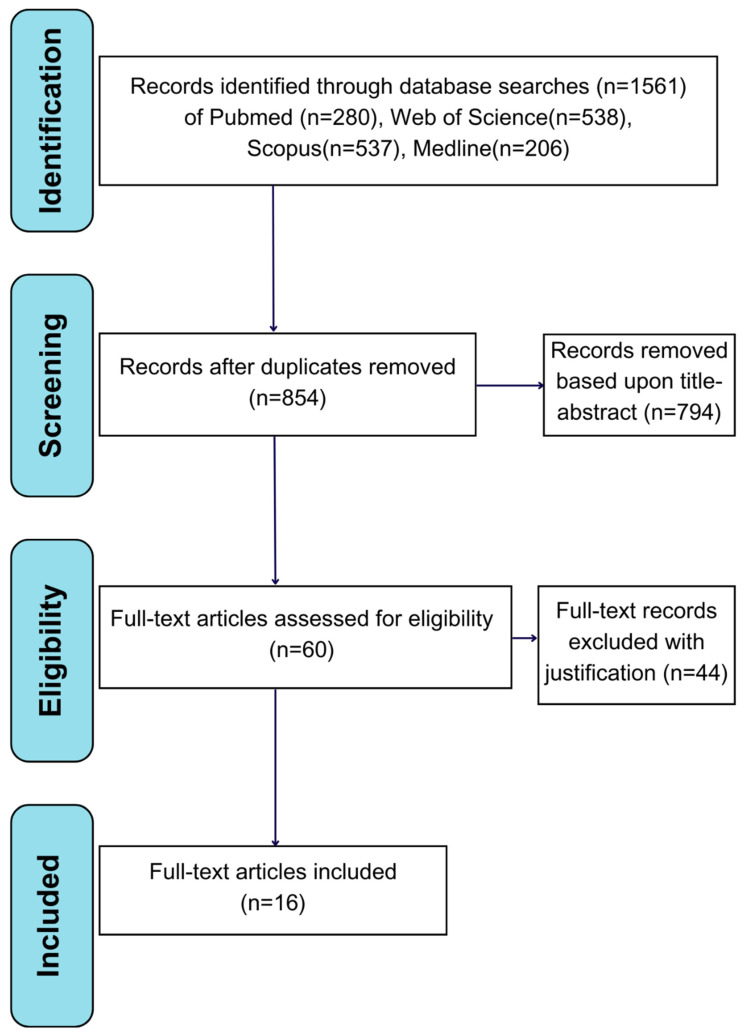
PRISMA flow diagram, adapted from [25].

**Table 1 ijms-26-09330-t001:** General characteristics of studies.

Reference	Brief Summary	ADSC Source	Total Number of Subjects (*n*)	ADSC Culture Conditions
Aras et al. (2016) [32]	Rat-derived ADSCs were implanted into the site of SCI, and the effects on locomotor recovery measured in rats.	Rat-derived ADSCs	42 + 6 sacrificed during study	Harvested adipose tissue digested in DMEM, filtered and then centrifuged at 1200 rpm for 10 min. Medium replaced with fresh culture after 7 days then biweekly thereafter.
Ohta et al. (2017) [33]	ADSCs were harvested from the dorsal fat pads of rats, infused into rats following SCI and the impact upon locomotor recovery was measured.	Rat-derived ADSCs Dorsal fat pad	30	Centrifuge products suspended in medium at 37 °C in atmosphere of 5% CO_2_ 95% air. Non-adherent cells removed after 24 h.
Zhou et al. (2013) [30]	Compared the effects of ADSC and BMSC implantation on functional recovery in rat models following SCI.	Human-derived ADSCs (bone-marrow cells also included)	108	ADSCs obtained following centrifuging process were suspended in DMEM with 10% FBS. Non-adherent cells removed by PBS wash.
Sarveazad et al. (2017) [34]	ADSCs were harvested from human tissue and implanted into rats alongside chondroitinase ABC, subsequent impact on functional recovery following SCI was measured.	Human-derived ADSCs	30	Isolated ADSCs suspended in medium (DMEM, HAM’s F-12, 10% FBS, 1% P/S). Cells incubated for 4 passages.
Takahara et al. (2024) [28]	ADSCs derived from rats were implanted in the subpial region, the impact upon functional recovery following aortic occlusion/reperfusion-induced SCI was measured in rats.	Rat-derived ADSCs	12	ADSCs cultured in DMEM with 10% FBS. Passage 0 ADSCs washed with PBS after 24 h initial incubation. Maintained in DMEM, 10% FBS and 1% P/S. Medium changed every other day.
Zhang et al. (2009) [27]	Differentiated and undifferentiated ADSCs were implanted into rats following SCI, with the relative impacts on functional recovery compared.	Rat-derived ADSCs	46	ADSCs obtained following centrifuging process maintained in DMEM/F12, 10% FBS, 1% P/S. Cultures incubated at 37 °C with 5% CO_2_.
Yousefifard et al. (2022) [31]	Human-derived ADSCs were implanted into rats following SCI, the impacts on functional recovery and pain following SCI were measured.	Human-derived ADSCs	42 Excluded animals with BBB > 3 following SCI	ADSCS obtained following centrifugation process cultured in DMEM, Ham SF-12, 10% FBS, 1% P/S. Cells incubated until third passage.
Ryu et al. (2009) [29]	Canine-derived ADSCs were transplanted into dogs following SCI, the impacts on functional recovery were measured.	Canine-derived ADSCs from 2-year-old dog	11	ADSC pellets obtained after centrifuging cultured in DMEM, 10% FBS. Unattached cells washed off with PBS. Medium supplemented with rEGF, bovine pituitary extract, 2 mM N-acetyl-L-cysteine, 0.2 mM L-ascorbic acid, 2-phosphate, insulin and hydrocortisone. Medium changed at 48 h intervals until cell confluence was achieved.
Takahashi et al. (2023) [35]	Rat-derived ADSCs were implanted into rats following severe SCI, the impacts on functional recovery and the mechanisms that subserve this were examined.	Rat-derived ADSCs	80	Cells washed with PBS, cultured in DMEM and 10% FBS. Cultures maintained at 80–90% confluency at 37.5 °C with 5% CO_2_.
Min et al. (2017) [36]	Rat-derived ADSCs were infused into SCI rats alongside GCSF, the impacts upon functional recovery were measured.	Rat-derived ADSCs	28	Pellet obtained following centrifuging incubated at 37 °C in 5% CO_2_. Non-adherent cells removed by replacing medium; cells fed every 3 days by feeding with DMEM solution.
Sarveazad et al. (2014) [37]	Human-derived ADSC therapy and chondroitinase ABC therapy were compared in terms of their impacts on functional recovery following SCI in rat models.	Human-derived ADSCs	24	Formed plate following centrifuging process cultured in DMEM/Ham’s F-12, 10% FBS and 1% P/S. Flasks incubated at 37 °C, 5% CO_2_ and 98% humidity.
Junior et al. (2020) [38]	Human-derived ADSCs were implanted into rats following SCI, the impacts on locomotor recovery and bladder function were measured.	Human-derived ADSCs	63	Details of culture conditions unclear.
Li et al. (2022) [39]	Human-derived ADSCs and nano hydrogels were implanted into rats following SCI, the impact upon functional recovery was then measured.	Human-derived ADSCs	32	Cells cultured in ADSC complete medium. Culture changed one day after seeding, then every 3 days thereafter, for a total culture time of 7 days.
Zhang et al. (2024) [40]	Rat-derived ADSCs were implanted into SCI rats, the impact on functional recovery and PGRN expression was monitored.	Rat-derived ADSCs	30	Primary ADSCs cultured in DMEM, 10% FBS and 1% P/S.
Li et al. (2021) [41]	Human-derived ADSCs were implanted into rats following SCI, the impacts upon locomotor recovery and relevant molecular pathways were investigated.	Human-derived ADSCs	30	ADSCs cultured in alpha-MEM, 37 °C and 5% CO_2_.
Oh et al. (2012) [42]	Human-derived ADSC clusters were implanted into rats following SCI, the impact on locomotor recovery was examined.	Human-derived ADSCs	30	Cells cultured in DMEM/F-12. Plated in tissue flasks in culture conditions of humidified air at 37 °C 5% CO_2_ and 95% air. Culture changed to remove non-adherent cells, and media changed every other day.

DMEM = Dulbecco’s Modified Eagle Medium; BMSC = bone-marrow derived stem cells; FBS = Foetal Bovine Serum; P/S = Penicillin/Streptomycin; PBS = Phosphate-Buffered Saline; Alpha-MEM = alpha Minimum Essential Medium.

**Table 2 ijms-26-09330-t002:** Results of ADSC characterisation.

References	Characterisation Method	Results
Aras et al. (2016) [32]	Flow cytometry	CD29, CD54 and CD90 expressed in rat-derived ADSCs.
Ohta et al. (2017) [33]	Flow cytometry and mRNA characterisation (RT-PCR)	Beta-3 tubulin and GFAP expression not significantly altered with regard to ADSC passage number. Significant reduction in nestin expression with regards to increasing passage number.
Zhou et al. (2013) [30]	Flowcytometry, immunocytochemistry and RT-PCR	CD29, CD44, CD90 and CD105 expressed in ADSCs. Cells positive for vimentin and BDNF. Nestin expressed in ADSCs at 10.25% ± 4.5%; Ki67 expressed in ADSCs at 57.22% ± 5.76%.
Sarveazad et al. (2017) [34]	Flow cytometry	CD44, CD73 and CD90 expressed in ADSCs.
Takahara et al. (2024) [28]	Flow cytometry	CD29 and CD90 expressed in ADSCs.
Zhang et al. (2009) [27]	Immunocytochemistry	Nestin, vimentin, CD44 and beta-tubulin III expressed in undifferentiated ADSCs.
Yousefifard et al. (2022) [31]	Flow cytometry	CD44, CD73 and CD166 expressed in ADSCs.
Ryu et al. (2009) [29]	Flow cytometry	First passage ADSCs expressed CD44, CD90, CD105 and MHC class I; partially positive for CD34. Seventh passage of ASSCs expressed CD44, CD90 and CD105.
Takahashi et al. (2023) [35]	Flow cytometry	Third passage ADSCs positive CD29 and CD90.
Min et al. (2017) [36]	Flow cytometry	ADSCs expressed CD73 and CD105 at rates of 90.1% and 98.6%.
Sarveazad et al. (2014) [37]	Flow cytometry	CD44, CD73 and CD90 expressed in ADSCs.
Junior et al. (2020) [38]	Immunophenotyping	CD29, CD73, CD90 and CD105 expressed in ADSCs.
Li et al. (2022) [39]	Flow cytometry	Fifth generation ADSCs expressed CD73, CD90 and CD105 > 95%.
Zhang et al. (2024) [40]	Flow cytometry	CD44 and CD90 expressed by ADSCs.
Li et al. (2021) [41]	Flow cytometry	ADSCs expressed high levels of CD13, CD44, CD73, CD90 and CD105.
Oh et al. (2012) [42]	Immunofluorescence	ADSCs between passages 3–5 expressed human CD29, CD90 and CD105.

CD = cluster of differentiation; RT-PCR = reverse transcriptase polymerase chain reaction.

**Table 3 ijms-26-09330-t003:** SCI experimental models.

References	SCI Injury Level	SCI Injury Method
Aras et al. (2016) [32]	T10–T11	Severe SCI trauma.
Ohta et al. (2017) [33]	T10 exposed by laminectomy	Weight-drop method: 10 g weight from 25 mm height onto exposed spinal cord.
Zhou et al. (2013) [30]	Bilateral T9 exposure by dorsal laminectomy	CST, RST and dorsal columns cut with microscissors to depth of central canal.
Sarveazad et al. (2017) [34]	T8–T9 exposure by laminectomy	Incision and piston method, causing moderate contusion SCI. 10 g piston released from 12.5 cm distance.
Takahara et al. (2024) [28]	SCI induced by aortic occlusion/reperfusion leading to infarction [multi-level injury]	TairaMarsala technique: 2F Fogarty catheter introduced through left femoral artery for aortic occlusion. 20G catheter in left common carotid artery for blood withdrawal. Blood pressure maintained at 40 mmHg during 10 min occlusion period.
Zhang et al. (2009) [27]	T9–T10 exposure by laminectomy	Weight-drop method: 10 g rod, 2.5 mm in diameter dropped from 12.5 mm height.
Yousefifard et al. (2022) [31]	T6–T8 exposure by laminectomy	Moderate clip compression method: 60 s aneurysm clip compression at pressure 20 g/cm^2^.
Ryu et al. (2009) [29]	L4 exposure by hemilaminectomy	Epidural balloon catheter method, 12 h injury period.
Takahashi et al. (2023) [35]	T9–T10 exposure by total laminectomy	Contusion SCI: ACI device at 250 kD leading to severe SCI.
Min et al. (2017) [36]	T7 [bilateral dorsal laminectomy]	Contusion SCI: infinite horizon impactor device with 200 kdyn force.
Sarveazad et al. (2014) [37]	T8–T9 exposure by laminectomy	Weight drop; 10 g 2 mm diameter cylinder released from 12.5 cm distance.
Junior et al. (2020) [38]	Thoracolumbar region	Balloon-induced compressive SCI; filled with 80 microlitres saline.
Li et al. (2022) [39]	T10 exposure by laminectomy	Piston-induced SCI; 2.2 mm diameter impactor at 1.4 mm impact depth and 0.6 s residence time.
Zhang et al. (2024) [40]	T10 exposure	Contusional SCI; 10 g weight drop from 2.5 cm height.
Li et al. (2021) [41]	T10 exposure by laminectomy	Bulldog clamp method; 30 s injury period.
Oh et al. (2012) [42]	T9 exposure by laminectomy	Clip compression SCI; 10 min injury period.

CST = corticospinal tract; RST = rubrospinal tract; ACI = Automated Computer-controlled impactor.

**Table 4 ijms-26-09330-t004:** Locomotor recovery outcomes.

Reference	Experimental Groups	Results
Aras et al. (2016) [32]	L + T L + T + PSH/Hyperacute L + T + MSC/Hyperacute L + T and L + T + PSH/Acute L + T & L + T + MSC/Acute	MSC therapy consistently resulted in significant difference in BBB scores overall. Significant differences found between following groups: L + T//L [*p* = 0.004 **] L + T//L + T + MSC/Hyperacute [*p* = 0.007 **] L + T//& L + T + MSC/Acute [*p* = 0.017 *] L + T + PSH/Hyperacute//L + T + MSC/Hyperacute [*p* = 0.0010 ***] L + T + PSH/Acute//L + T + MSC/Acute [*p* = 0.0014 **] No significant differences between L + T and L + T + PSH [both acute/hyperacute].
Ohta et al. (2017) [33]	Saline treatment only ADSC therapy	Significantly higher BBB scores in ADSC group [all timepoints beyond 8 days, point of intervention]. *p* < 0.01 [*] at all timepoints therein, except 11-, 42-, 50-day timepoints [*p* < 0.05 [*]].
Zhou et al. (2013) [30]	PBS control hBMSC hADSC	Significantly higher BBB scores in hADSC score vs. Control [all timepoints beyond 2 weeks, [*p* < 0.05 [*]]. Average BBB score of 12.5 in ADSC group, significantly higher than other 2 groups. No other significant differences between groups at any timepoint [not hADSC vs. hBMSC nor hBMSC vs. control]. One-way analysis of variance followed by Bonferroni post hoc testing.
Sarveazad et al. (2017) [34]	Sham SCI only hADSC ChABC hADSC + ChABC	Single treatment groups had significantly higher BBB scores compared to SCI only [all timepoints beyond 28 d in hADSC group; 14,21 + 63 d timepoints in ChABC group; *p* < 0.001]. Dual therapy group had significantly higher BBB scores compared to SCI only [all timepoints beyond 14 d] + compared to single treatment groups [all timepoints beyond 21 d]. [*p* < 0.001]. Treatment is superior to control, and dual treatment is superior to single therapy with respect to locomotor recovery.
Takahara et al. (2024) [28]	Control [PBS] ADSC	Significantly higher BBB scores in ADSC group vs. Control days 7 + 14 post-treatment [*p* < 0.05].
Zhang et al. (2009) [27]	Sham control Saline control dADSC-P1 dADSC-P2 uADSC	Significant improvement in BBB score between cell-treated vs. Control groups at end of 12 weeks: *p* < 0.01: uADSC, dADSC-P2 *p* < 0.05: dADSC-P1 Significant difference between uADSC//dADSC-P2, dADSC-P1 groups [*p* < 0.05 both cases]. No other significant differences [recovery timecourses of uADSC//dADSC-P2 groups].
Yousefifard et al. (2022) [31]	Intact Sham SCI only Vehicle [cell culture media] ADSC	Significantly improved BBB scores [ADSC group vs. SCI group, all timepoints beyond 4 weeks, *p* < 0.0001].
Ryu et al. (2009) [29]	Control [SCI only] Vehicle [PBS] ADSC	Significantly higher Obly [modified BBB scores [ADSC group vs. Other groups, 5 + 9-week timepoints; *p* < 0.05]. ADSC group Olby scores increased 1 [3 W], 3.6 [5 W] and 4.6 [9 W]. Other groups consistently displayed Olby scores < 1.
Takahashi et al. (2023) [35]	Sham PBS no exercise PBS exercise ADSC no exercise ADSC exercise	Significantly higher BBB scores [ADSC-ex vs. All other groups at 7, 8, 10 weeks post-transplant; *p* < 0.05]. No other significant differences between groups, including ADSC no exercise vs. PBS [± exercise].
Min et al. (2017) [36]	Sham GCSF ADSC ADSC + GSCF	At 8 W: ADSC, ADSC + GCSF groups had mean BBB scores of 17.37 ± 0.7 and 17.57 ± 0.5; significantly higher than GCSF, Sham; 16 ± 0.5 and 15.6 ± 0.5, respectively [*p* < 0.01]. No significant difference [ADSC//ADSC + GCSF groups nor GCSF//Control].
Sarveazad et al. (2014) [37]	Control Sham ChABC hADSC	Significant difference between hADSCs//Control] 28 d, *p* < 0.01]. Significant difference in BBB score between Ch ABC//control [14 d, *p* < 0.001]. Significant difference between Ch ABC, hADSCs//Control [63 d, *p* < 0.01]. Intervention point at 7 days post-surgery.
Junior et al. (2020) [38]	Control hADSC hADSC + MPSS	Motor recovery at 21 d [hADSC] + 17 d [hADSC + MPSS] following second transplant. Median BBB score following motor stabilisation: 5.5 [hADSC] and 5.0 [hADSC + MPSS]; no subsequent regression of motor assessment. Significantly improved motor function [hADSC ± MPSS//Control]. *p* < 0.01.
Li et al. (2022) [39]	PBS ADSC RADA16-RGD ADSC + RADA16-RGD	Significantly higher BBB scores [ADSC + RADA16-RGD//All other groups] between 3 and 10 weeks. *p* values: ADSCs + RADA16-RGD//PBS: *p* < 0.01 at 4, 5, and 7 w; *p* < 0.001 after 3, 6, 8, 9, and 10 w. ADSCs + RADA16-RGD//ADSCs: *p* < 0.01 after 4 and 7 w; *p* < 0.001 after 3, 5, 6, 8, 9, and 10 w. ADSCs + RADA16-RGD//RADA16-RGD: *p* < 0.05 after 3, 4, 6, and 7 w; *p* < 0.01 after 8, 9, and 10 w.
Zhang et al. (2024) [40]	Sham SCI alone SCI + PBS SCI + ADSCs	Significantly higher BBB score in SCI + ADSC//SCI [*p* < 0.05 at 8 w, *p* < 0.01 at 9, 10 W]. Significantly higher BBB score in SCI + ADSC//SCI + PBS [*p* < 0.05 at 8, 10 W; *p* < 0.001 at 9 W].
Li et al. (2021) [41]	Sham PBS SB ADSC ADSC + SB	Significant difference in BBB score [ADSC vs. Baseline at 3 d; at 7, 14 d, *p* < 0.05]. Significant difference in BBB score [ADSC + SB//Baseline at 3 d; 14, 21 d, *p* < 0.05]. Significant difference in BBB score [ADSC//ADSC + SB; at 28 d, *p* < 0.05].
Oh et al. (2012) [42]	PBS 3DCM hADSC	Significant difference in BBB scores between 3DCM//hADSC, 3DCM//PBS, hADSC//3DCM beyond 3 W [*p* < 0.05]. Significant difference between 3DCM//hADSC, not hADSC//PBS at 1, 2 W [*p* < 0.05].

Notation: Second column contains the experimental groups (separated by spaces). + denotes a group that receives both interventions. / denotes the administration time in the first row (hyperacute or acute phase of SCI). // denotes groups that are being compared to derive stated *p* values, e.g., hADSC//PBS = hADSC compared to PBS. BBB = Basso, Beattie, Bresnahan; L + T = laminectomy + trauma; PSH = physiological saline; chABC = chondroitinase ABC; dADSC = differentiated ADSCs; uADSC = undifferentiated ADSCs; GSCF = granulocyte-colony stimulating factor; RADA16-RGD = self-assembling peptide; SB = TGF-1R inhibito; 3-DCM = three-dimensional cell mass. *p* value notation: * *p*, 0.01 < *p* < 0.05; ** *p*, 0.001 < *p* < 0.01; *** *p*, *p* < 0.001.

**Table 5 ijms-26-09330-t005:** Bladder recovery.

Reference	Summary of Results	Significance
Junior et al. (2020) [38]	Reddish urine occurred in 43/63 (68.7%) rats for three days following SCI. Two animals (one from group A, one from B) urinary retention not responsive to bladder massage; cystocentesis resulted in death. Urinary continence recovery not observed in animals group A (control). Groups B (ADSC) and C (ADSC + MPSS) urinary continence recovery rates of 14/21 (66.66%) and 13/21 (61.9%); average of 9 days from the second transplant. Significant difference between A and B regarding urinary continence [*p* < 0.01], no significant difference between groups B and C [*p* < 0.0657].	Evidence that ADSC therapy promotes statistically significant increases recovery of bladder function following SCI, as compared to control. No evidence that adjuvant therapy with MPSS yields additive effects with respect to ADSC therapy in terms of bladder function.
Zhang et al. (2024) [40]	Bladder histology examined following SCI. ADSC group showed less significant histological damage to bladder tissue (HE + Masson staining). ADSC groups displayed lesser degree of following histological changes: epithelial layer of bladder sphincter mucosa thickening, mucosal epithelial shedding, inflammatory cell infiltrate (lamina propria), elastic fibre reduction, collagen deposition, enlarged smooth muscle cell nuclei, muscle cell disorder and bladder stone formation.	Indication ADSC therapy reduces extent of damage following SCI with respect to bladder histology. Likely translates to improved bladder function. Statistical significance not quantified. Histology less severe (ADSC group), not identical to sham.

MPSS = methylprednisolone sodium succinate.

**Table 6 ijms-26-09330-t006:** Risk of bias assessment.

	Study ID															
Category	1	2	3	4	5	6	7	8	9	10	11	12	13	14	15	16
Randomised Sequence generation	2	2	2	2	2	2	2	2	2	2	2	2	2	2	2	2
Baseline characteristics	2	1	3	2	2	2	2	2	2	1	2	2	1	2	3	2
Allocation concealment	N/A	N/A	N/A	N/A	N/A	N/A	N/A	N/A	N/A	N/A	N/A	N/A	N/A	N/A	N/A	N/A
Random housing	2	2	2	2	2	2	2	2	2	1	1	2	2	1	2	2
Blinding of allocation	1	2	1	1	2	2	2	1	1	1	1	1	1	1	2	2
Random outcome assessment	2	2	2	2	2	2	1	1	1	1	1	1	1	1	2	2
Blinding of assessment	1	2	1	1	2	2	2	1	1	1	1	1	1	1	2	2
Attrition bias	1	2	2	2	1	1	1	2	2	1	3	2	3	2	3	2
Reporting bias	2	2	1	1	1	1	1	1	2	2	2	2	2	2	2	2
Other sources of bias	1	1	1	1	1	1	1	1	1	1	1	1	1	1	2	2

1 = Aras et al. (2016) [32]; 2 = Ohta et al. (2017) [33]; 3 = Zhou et al. (2013) [30]; 4 = Sarveazad et al. (2017) [34]; 5 = Takahara et al. (2024) [28]; 6 = Zhang et al. (2009) [27]; 7 = Yousefifard et al. (2022) [31]; 8 = Ryu et al. (2009) [29]; 9 = Takahashi et al. (2023) [35]; 10 = Min et al. (2017) [36]; 11 = Sarveazad et al. (2014) [37]; 12 = Junior et al. (2020) [38]; 13 = Li et al. (2022) [39]; 14 = Zhang et al. (2024) [40]; 15 = Li et al. (2021) [41]; 16 = Oh et al. (2012) [42].

## Data Availability

Not applicable.

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
