# Peer review of "Adipocyte-Derived Stem Cells in the Treatment of Spinal Cord Injuries in Animal Models: A Systematic Review"

_ijms, 2025, doi:10.3390/ijms26199330_

Round 1
Reviewer 1 Report
Comments and Suggestions for Authors
Authors evaluated 16 studies on using adipocyte-derived stem cells (ADSCs) for treating spinal cord injuries in animal models. The analysis covers ADSC characterization, injury models, and functional recovery outcomes. While ADSC therapy shows promise in improving motor function, the evidence for adjuvant therapies is variable. Comments,
- Can you elaborate on the heterogeneity of ADSC characterization methods across studies?
- What measures were taken to ensure consistency in assessing study quality?
- How might the species of ADSC origin affect translatability to humans?
- Could you explain the rationale behind the chosen sample size variations?
- How generalizable are the findings given the lack of cervical SCI models?
- What steps were taken to minimize bias in outcome assessment?
- How does the timing of ADSC administration impact therapeutic efficacy?
- Could you discuss limitations in extrapolating results from thoracic to cervical injuries?
- How might age-related factors influence the applicability of these findings?
- What are the implications of using different injury induction methods?
- How reliable are BBB scores as the sole measure of functional recovery?
- Can you comment on the potential long-term effects not captured in these studies?
- How might comorbid conditions affect ADSC therapy outcomes?
- How could future studies address the gaps in understanding mechanisms of action?
Author Response
Comment 1:
Can you elaborate on the heterogeneity of ADSC characterization methods across studies?
Response 1:
Dear esteemed reviewer, thank you for your question. 11 of 16 studies utilised flow cytometry alone, 2 combined flow cytometry with another method, and the remaining 3 used alternate methods. This heterogeneity of approaches may partly account for the heterogeneity in the results of ADSC characterisation. We draw particular attention to the 2 studies that utilised RT-PCR alongside flow cytometry, as RT-PCR can detect intracellular mRNA expression. We recommend further studies adopt the use of RT-PCR alongside flow cytometry to better characterise ADSCs beyond surface markers alone. We thank you for drawing our attention to this point.
Comment 2:
What measures were taken to ensure consistency in assessing study quality?
Response 2:
Dear esteemed reviewer, thank you for your question. A modified version of SYRCLE’s ROB tool was applied by each reviewer independently, and any discrepancies were solved by discussion with a third independent reviewer.
Comment 3:
How might the species of ADSC origin affect translatability to humans?
Response 3:
Dear esteemed reviewer, thank you for your question. Rat ADSCs and human ADSCs have some degree of shared functionality, although human ADSC studies are likely to display a greater degree of translational relevance to humans. 8 of 16 studies use human ADSCs as the source, we have made a note of this in the introduction, as well as the limitations of results section,4.7, that we have added to our study
Comment 4:
Could you explain the rationale behind the chosen sample size variations?
Response 4:
Dear esteemed reviewer, we thank you for your question. We excluded studies with fewer than 10 animals to minimize the risk of bias and increase the reliability of pooled findings. Excessively small sample sizes in preclinical studies can contribute to high variability, reduced statistical power, and greater variance due to random effects. Setting a threshold of ≥10 subjects ensures that included studies provide more robust and generalizable data, therefore improving the translational relevance of results. Sample sizes varied from 11 (canine study) to 108 (rat study). Small sample sizes in the canine study likely relates to cost and ethics-related considerations of using canines in preclinical research, compared with the relative ease and availability of rats.
Comment 5:
How generalizable are the findings given the lack of cervical SCI models?
Response 5:
Dear esteemed reviewer, thank you for your question. The lack of cervical SCI models is one of our primary concerns with the current evidence-base, especially given the relevance of cervical SCI to human patterns of dysfunction. To reflect this, we have added a discussion to section 4.7 emphasising the need for more cervical SCI models in the literature. We thank you again for raising this critical point.
Comment 6:
What steps were taken to minimize bias in outcome assessment?
Response 6:
Dear esteemed reviewer, thank you for your question. Unfortunately, most literature on the subject fails to explicitly describe outcome assessment protocols, beyond the mentioning of blinding reviewers. We have added a sentence in section 4.7 to address this weakness in the literature, and implore greater methodological rigour in this aspect.
Comment 7:
How does the timing of ADSC administration impact therapeutic efficacy?
Response 7:
Dear esteemed reviewer, thank you for your question. From the evidence presented, we believe ADSC therapy to be most effective when administered during the acute phase of SCI, as there is evidence for molecular protection mechanisms through cellular support provided by ADSC cells. However, one pattern we noted is the observed time lag for ADSC therapy to take effect, the exact basis of this is uncertain. We thank you again for drawing this to light.
Comment 8:
Could you discuss limitations in extrapolating results from thoracic to cervical injuries?
Response 8:
Dear esteemed reviewer, thank you for your question. The primary limitation is that thoracic and cervical spinal injuries differ in terms of neurological dysfunction, with cervical injuries being the cause of tragic quadriplegic phenotypes. We have added discussion of these limitations to section 4.7.
Comment 9:
How might age-related factors influence the applicability of these findings?
Response 9:
Dear esteemed reviewer, thank you for your question. This question is an interesting one given the relevance of age in human SCI patients. We believe a potential outcome is that age limits the efficacy of ADSC therapy, due to the general biological depletion and slowing of processes that occurs with age. Unfortunately, none of the reviewed studies investigate how age influences outcomes. Whether human ageing can be accurately recapitulated in mice is a further debate, which we would welcome exploration of within the field. We have added a note to section 4.3 recommending the development of aged mice models to further investigate this idea. We thank you again for this most pertinent comment.
Comment 10:
What are the implications of using different injury induction methods?
Response 10:
Dear esteemed reviewer, we thank you for your comment. There is clear variance in the method of injury. There are two implications, first is this limits comparison between studies to some extent. Moreover, we’ve added further caveat to section 4.3 to address that experimental injury methods do not accurately mimic human patterns of SCI. Human spinal cords are protected by vertebrae, whilst the animal models in this study have their spine directly exposed. We think this is a key weakness of the current literature and implore further innovation to address this.
Comment 11:
How reliable are BBB scores as the sole measure of functional recovery?
Response 11:
Dear reviewer, we thank you for raising this point. The BBB scoring scale is widely used in rodent studies for assessing locomotor recovery, with the Olby scale being a modified variant suitable for canine subjects. There are, of course, facets of motor function beyond the BBB system, especially as these studies focus on hindlimb function. Moreover, rats lack the fine motor control of humans, so functional recovery measurements in mice are inherently limited with regards to translational relevance. Still, hindlimb function is an arguable proxy for lower limb function, and hence locomotor capability. We have added a note to section 4.7 to address the limitations of the BBB scoring system and its impact on our results. Thank you for raising this point.
Comment 12:
Can you comment on the potential long-term effects not captured in these studies?
Response 12:
Dear esteemed reviewer, we thank you for your comment. Indeed these studies focus on a short-to-medium term outlook of ADSC therapy. Potential long-term effects are likely to centre around how ADSC cells integrate and function into the body. There may be further benefits, such as increased tissue generation potential and a reserve of cells that retain stemness. However, deleterious effects are also possible. Stem cells do have superior differentiation capacity, and whether this promotes the incidence of tumours, for example, is unclear for these results. We thank you again for bringing this to light.
Comment 13:
How might comorbid conditions affect ADSC therapy outcomes?
Response 13:
Dear esteemed reviewer, we thank you for your question. Comorbid conditions may influence the efficacy of ADSC therapy, particularly in the context of inflammatory conditions as this influences overall cytokine profiles, which may influence recovery following SCI. We think this is an important aspect of note, and have therefore added a note to section 4.4. Many human patients do suffer from several conditions at once, and investigating ADSC therapy in rodent models which aim to recapitulate this would be most wise.
Comment 14:
How could future studies address the gaps in understanding mechanisms of action?
Response 14:
Dear esteemed reviewer, we thank you for your question. Understanding the mechanisms of ADSC therapy is crucial in assessing whether this is a feasible strategy in humans, and how to best optimise said strategy. It is also incredibly complex given the sheer range of potential mechanisms, from cytokines to direct cell-level interactions, which may contribute. We suggest addressing these gaps will require both intelligent use of current technology, e.g. combining ADSC therapy with pathway-specific inhibitors, and the development of novel technology. We have added a note to our conclusion to reflect this, molecular pathway dissection is incredibly challenging and we sincerely hope these mechanisms are further elucidated in future.
Reviewer 2 Report
Comments and Suggestions for Authors
Please see the attached file. Thank you!

Author Response
Reviewer's preface:
The present review evaluates the potential of ADSC therapy as a strategy for treating SCI in animal models. A total of 1,561 studies were initially identified through PubMed, Web of Science, Scopus, and Medline. Following the application of predefined inclusion and exclusion criteria, 16 articles were ultimately included for detailed assessment and reporting. Overall, the manuscript is well written, logically structured, and clearly presented. However, there are several important issues that need to be addressed before the manuscript can be considered for publication.
Comment 1:
Please double-check the reported numbers in Section 2.1 and Figure 1, as they currently appear to be inconsistent.
Response 1:
Thank you for drawing attention to this error, we apologise profusely and have corrected the numbers to ensure consistency and accuracy across section 2.1 and Figure 1. Changes in text are highlighted in red.
Comment 2:
The BBB scale should be introduced in the Introduction section and then described in greater detail in the Methods section, so that readers unfamiliar with the scale understand what it is and how it was applied in this study.
Response 2:
Thank you for noting this, we have added a sentence in the introduction to address this point and a paragraph (section 2.6) has now been added to the methods to set out the scoring system’s principles.
Comment 3:
The authors should briefly describe the animal models used in the 16 selected studies.
Response 3:
Thank you for noting this, we have added a brief description of the animal models used in our study to the introduction section.
Comment 4:
Please spell out PRISMA in full when it first appears in the abstract, as abbreviations should be defined upon first mention. There are several similar cases throughout the manuscript; please revise accordingly.
Response 4:
Thank you for drawing attention to this, we have spelled out PRISMA in the abstract and revised further acronyms throughout the review as appropriate.
Comment 5:
The keywords should be revised. Please use the full names of all abbreviations, and note that the BBB scoring system is not mentioned in the abstract.
Response 5:
Thank you for noting this, we have revised the key words accordingly and removed the BBB scoring system from the keywords list.
Reviewer 3 Report
Comments and Suggestions for Authors
1. Novelty and Rationale
-
The manuscript does not sufficiently emphasize the specific knowledge gap addressed compared to other existing reviews on ADSCs for SCI.
-
How does this synthesis provide added value beyond summarizing the current literature?
2. Search Strategy and Methodology
-
The PRISMA-based search is described but lacks granularity (exact search terms, Boolean operators, time frame).
-
The exclusion of studies with n < 10 seems arbitrary; how does this threshold affect the comprehensiveness of the review?
-
No discussion on how language restrictions or database coverage may have influenced the retrieved studies.
3. Heterogeneity of Included Studies
-
Animal models vary substantially (contusion, compression, ischemia), making direct comparisons difficult.
-
The authors should address how such biological heterogeneity impacts the validity of pooled interpretations.
-
Is there any attempt to stratify results according to injury model, severity, or species?
4. Characterization of ADSCs
-
Surface marker panels differ significantly between studies (e.g., CD73, CD90, CD105 ± others).
-
The absence of a standardized marker set reduces comparability. What minimal set of markers would the authors recommend for uniform characterization?
5. Functional Outcomes
-
Most studies report improvements in BBB score, but potential publication bias is not acknowledged.
-
Could a quantitative meta-analysis (forest plot, pooled effect size) be feasible, at least for locomotor outcomes?
-
The variability in functional testing protocols across studies weakens cross-study comparisons.
6. Adjunctive Interventions and Synergy
-
The section on combinatory approaches (e.g., ADSCs + ChABC, exercise, G-CSF) lacks mechanistic interpretation.
-
Why do some combinations show additive effects while others do not?
-
Are there identifiable molecular pathways consistently targeted across synergistic interventions?
7. Functional Domains Beyond Locomotion
-
Only two studies assess bladder function, an endpoint of high translational relevance.
-
Why have autonomic functions (e.g., cardiovascular, bowel control) been largely neglected in the reviewed studies?
8. Translational Relevance
-
Most included studies use rodent thoracic models. Should future studies shift toward clinically relevant models (cervical SCI, large animals)?
-
Would organoid or “spinal cord-on-chip” models offer a more predictive preclinical bridge?
9. Risk of Bias and Quality of Evidence
-
Many studies scored “unclear” in risk of bias assessment.
-
Without stricter methodological reporting, how reliable are the observed positive effects?
-
The narrative would benefit from a stronger critical appraisal of the evidence base.
10. Conclusions
-
The manuscript ends on an overly optimistic note given the variability and methodological weaknesses of the included studies.
-
A more balanced conclusion highlighting both potential and uncertainty would increase credibility.
Author Response
Comment section 1:
Novelty and Rationale. The manuscript does not sufficiently emphasize the specific knowledge gap addressed compared to other existing reviews on ADSCs for SCI. How does this synthesis provide added value beyond summarizing the current literature?
Response section 1:
Dear reviewer, thank you for your comment. We have added a clear statement in the introduction to address the knowledge gap this paper fulfills. Namely, a systematic review investigating the use of undifferentiated ADSCs in animal models to improve locomotor recovery following SCI.
Dear esteemed reviewer, thank you for your question. This systematic review pools together results from animal studies investigating undifferentiated ADSCs and their efficacy in locomotor recovery following SCI, the first systematic review to do so. It also systematically engages with the current literature, and suggests potential molecular explanations, and dire changes required within the field. The current evidence-base requires stronger reporting and methodology standards and we feel this review supports this goal by highlighting this in our added criticisms of studies section (4.7).
Comment section 2:
Search Strategy and Methodology. The PRISMA-based search is described but lacks granularity (exact search terms, Boolean operators, time frame). The exclusion of studies with n < 10 seems arbitrary; how does this threshold affect the comprehensiveness of the review?
Response section 2:
Dear esteemed reviewer, thank you for your comment. We have added more granularity to the PRISMA search strategy in response to your concern, as well as an exact time frame. No discussion on how language restrictions or database coverage may have influenced the retrieved studies.
Dear esteemed reviewer, we thank you for your question. We excluded studies with fewer than 10 participants because such sample sizes reduce the translatability of results. Very small studies are disproportionately affected by random variation, lack statistical power to detect reliable effects, and often represent preliminary or pilot work rather than robust evidence. Including them would risk inflating noise and bias, thereby reducing the validity of pooled conclusions. By applying a minimum threshold of n = 10, we ensured that all included studies had at least a basic level of statistical credibility and generalisability, thereby strengthening the reliability and interpretability of the review’s findings. Sample sizes below 10 may skew results away from normal distributions, making certain statistical approaches unfeasible. We feel this is especially important given some of the methodological weaknesses present in the ADSC research field, particularly in smaller studies and case reports. We thank you again for drawing this to our attention, as it highlights a broader need for the field to improve the quality of study designs.
Dear esteemed reviewer, we thank you for your comment. We have added a statement in the discussion under section 4.7 to address your observation here. We thank you for bringing this to light and have made this change accordingly.
Comment section 3:
Heterogeneity of Included Studies. Animal models vary substantially (contusion, compression, ischemia), making direct comparisons difficult. The authors should address how such biological heterogeneity impacts the validity of pooled interpretations. Is there any attempt to stratify results according to injury model, severity, or species?
Response section 3:
Dear reviewer, we thank you for your comment. We agree that the heterogeneity of animal models creates challenges in comparing results. To address your comment we have expanded commentary on the limitations of these results now added to section 4.7
Dear esteemed reviewer, we thank you for your question. In response to this we have added commentary in the discussion under section 4.4 discussing how results vary across animal models. We thank you for bringing this to our attention, and have altered our manuscript accordingly.
Comment section 4:
Characterization of ADSCs. Surface marker panels differ significantly between studies (e.g., CD73, CD90, CD105 ± others). The absence of a standardized marker set reduces comparability. What minimal set of markers would the authors recommend for uniform characterization?
Response section 4:
Dear reviewer, we thank you for your question. The challenge of characterising stem cells is most pertinent to this study, and surface markers can differ significantly based upon isolation and culture approaches. The exact nature of what defines stemness is still partially elucidated. If we were to suggest a minimum criteria, we’d suggest at least 2 markers from the following list: CD29, CD44, CD73, CD90, CD105. To fully suggest accurate minimum criteria I believe further direct lab work would be required. Uniform characterisation remains exquisitely challenging and we thank you for your thought-provoking question.
Comment section 5:
Functional Outcomes. Most studies report improvements in BBB score, but potential publication bias is not acknowledged. Could a quantitative meta-analysis (forest plot, pooled effect size) be feasible, at least for locomotor outcomes?
Response section 5:
Dear esteemed reviewer, we thank you for your comment. We have added a statement to section 4.7 regarding this point in response to your statement. Publication bias is potentially a factor which could skew results. The variability in functional testing protocols across studies weakens cross-study comparisons.
Dear esteemed reviewer, we thank you for your question. This was something we explored, but due to the available studies failing to provide raw data, beyond graphs, we were unable to do so. This brings to light an important point, a dire need for increased data transparency in the literature, and we have added commentary to both section 4.7 and the conclusion to reflect this, and address your question.
Dear esteemed reviewer, thank you for your comment. We believe the BBB scoring system provides a strong degree of standardised cross-study comparisons, but note the limitations due to differences in functional testing protocols. We have revised some of our commentary in section 4.4 to address this comment. We thank you again for bringing this to our attention.
Comment section 6:
Adjunctive Interventions and Synergy. The section on combinatory approaches (e.g., ADSCs + ChABC, exercise, G-CSF) lacks mechanistic interpretation. Why do some combinations show additive effects while others do not? Are there identifiable molecular pathways consistently targeted across synergistic interventions?
Response section 6:
Dear esteemed reviewer, we thank you for your question. We suggest this is due to synergistic pathway activation, in the case of additive effects, compared to parallel pathways that do not interact. The exact nature of these pathways and mechanisms is still to be fully elucidated, and we’ve added commentary to section 4.6 to further reflect this. Thank you again for highlighting this.
Dear esteemed reviewer, we thank you for your comment. We believe this question most pertinent, and a full understanding of the molecular pathways involved would require lab-based pathway dissection experiments. We propose some degree of involvement of cytokine receptors, particularly TGF-1R, but the full mechanistic picture is yet to be fully elucidated. We have added statements on these highlighted discussion points to section 4.6 to reflect a more mechanistic interpretation to combinatory approaches, and illustrating the need for further pathway investigation, to fully realise the potential of ADSC therapy fully.
Comment section 7:
Functional Domains Beyond Locomotion. Only two studies assess bladder function, an endpoint of high translational relevance. Why have autonomic functions (e.g., cardiovascular, bowel control) been largely neglected in the reviewed studies?
Response section 7:
Dear esteemed reviewer, we thank you for your question. We believe that future studies most certainly should address impacts on autonomic function. We believe that the neglect of this metric is largely due to the sheer complexity in the assessment of parameters such as bowel control, cardiovascular function, etc. The assessment of motor function has been standardized to straightforward scales such as the BBB locomotor scale, a far less invasive methodology in comparison to specialist autonomic measures, which is why we chose to focus on this for the sake of cross-study comparisons. We thank you for highlighting this to us.
Comment section 8:
Translational Relevance. Most included studies use rodent thoracic models. Should future studies shift toward clinically relevant models (cervical SCI, large animals)? Would organoid or “spinal cord-on-chip” models offer a more predictive preclinical bridge?
Response section 8:
Dear esteemed reviewer, we thank you for your question. We believe future studies should shift towards more clinically relevant models, especially those that mimic cervical SCI given the relevance to human injury patterns. We think your question most pertinent and have therefore added a note to our conclusion to highlight this need. We thank you again for drawing this to our attention.
Dear esteemed reviewer, we thank you for your question. We believe such models may offer some benefits over animal models, namely the ability to more accurately recapitulate the human spinal cord environment. Of course, mimicking organism-level dynamics is challenging in such models and so we suggest a multi-faceted approach that integrates both more accurate animal models (cervical, large animals) alongside organoid models to form a clearer overall evidence-base. We thank you for bringing this fascinating discussion to our attention.
Comment section 9:
Risk of Bias and Quality of Evidence. Many studies scored “unclear” in risk of bias assessment. Without stricter methodological reporting, how reliable are the observed positive effects? The narrative would benefit from a stronger critical appraisal of the evidence base.
Response section 9:
Dear esteemed reviewer, we thank you for your question. We have added more extensive critical commentary to section 4.7 to address your comment; we agree wholeheartedly with your comment. There's a strong requirement in the field for clearer reporting standards, particularly in terms of clarity surrounding randomisation methods. We’ve extensively revised the tone of our discussion and conclusion to reflect this reality, and we thank you for bringing this to light.
Comment section 10:
Conclusions. The manuscript ends on an overly optimistic note given the variability and methodological weaknesses of the included studies. A more balanced conclusion highlighting both potential and uncertainty would increase credibility.
Response section 10:
Dear esteemed reviewer, we thank you for your comment. We have extensively revised the tone of our conclusion and added more critical nuance, in order to directly address your comment. We thank you again for taking the time to review our work in such detail, and highly value your input.
Reviewer 4 Report
Comments and Suggestions for Authors
The manuscript presents a robust methodology, and the authors appropriately acknowledge the heterogeneity of the data. However, the discussion could be strengthened by a deeper exploration of the molecular mechanisms involved in tissue regeneration. In addition, the manuscript shows significant issues in layout and formatting, requiring an extensive revision. Particular attention should be given to the organization of topics and the presentation of tables. The notation used to report the results in Table 4 is unclear and should be described in greater detail to ensure proper interpretation.
Author Response
Comment:
The manuscript presents a robust methodology, and the authors appropriately acknowledge the heterogeneity of the data. However, the discussion could be strengthened by a deeper exploration of the molecular mechanisms involved in tissue regeneration. In addition, the manuscript shows significant issues in layout and formatting, requiring an extensive revision. Particular attention should be given to the organization of topics and the presentation of tables. The notation used to report the results in Table 4 is unclear and should be described in greater detail to ensure proper interpretation.
Response:
Thank you esteemed reviewer for your feedback. We have added further discussion regarding molecular mechanisms, and approaches to further this understanding to section 4.6. We have also clarified the notation under table 4. We have reorganised topics and the general formatting extensively to better align with the expectations of the IJMS journal. We thank you again for your feedback, and have implemented it directly into our revised manuscript. This is exemplified by the reformatted tables, and additional sections added to the discussion to improve its depth. We thank you again for taking the time to provide us with detailed feedback, and have found this process thoroughly informative.
Round 2
Reviewer 1 Report
Comments and Suggestions for Authors
Authors have addressed the comments to my satisfaction. No further corrections required.
Reviewer 4 Report
Comments and Suggestions for Authors
The authors have successfully addressed the suggested changes. The current version of the manuscript has been significantly improved. For the final submission, we recommend a careful review of the manuscript's layout and formatting details.